# Indexing Model Based on Vector Normalization Available for Value Engineering in Building Materials

**Jongsik Lee**

Department of Architectural Engineering, Songwon University, Gwangju 61756, Korea; jslee@songwon.ac.kr

**Abstract:** Value engineering is a method of selecting the optimum design by evaluating the value of the original design and the alternative design. However, if the function score and the cost score of the evaluation subject are indexed, the range of the function index, the cost index and the value index calculated according to the functional attribute and the cost attribute may vary. The client is confused in the decision-making process of selecting the optimum design because the calculation range of the function, cost and value scores are different according to the evaluation subject. The necessity of indexing the cost score and the function score has been constantly raised, but it has been regarded as a difficult problem. This study presents a model that can index the function, cost and value scores using vector normalization method. Additionally, by applying this study model to the case of selecting finishing materials for the office automation floor of a building, the consistency of the study model was verified.

**Keywords:** building materials; value engineering; multi-objective decision-making; indexing model

## 1. Introduction

Since Value Engineering (hereinafter, VE) was first introduced at the General Electric Company by Lawrence D. Miles in 1947, the development of VE technique has been increasing in modern-day discussion of construction projects. As a unique management tool, VE can play important roles to reduce unnecessary costs and is convinced as a basis for enhancement of investment return in construction through increasing competitiveness, providing better satisfaction and mitigate the globalization impact on industry [1]. VE concepts used have a main focus to reduce the cost of the project by simply recommending the other advanced replaceable materials which are locally available to improve the value of the project [2]. Studies on VE present an increasing recognition and application in a number of business sectors, including the construction sector. This development has also been followed by establishment of VE professional institution in several countries, such as SAVE International in the United States, and the Institute of Value Management in the United Kingdom, Australia, Hong Kong, Malaysia and several other countries [3]. The SAVE International, founded by Lawrence D. Miles, called value methodology a "Systematic and structured approach for improving projects, products and processes [4]." VE in the construction industry is a method of analyzing the functions of buildings and creating alternative design in order to achieve the necessary functions with life-cycle cost (hereinafter, LCC) [5]. A regular procurer in North America would normally expect savings in the order of 8 to 25%, with an outlying range between 3 to 30% when VE is implemented [6].

The primary goal of VE is to improve project value.

A simple way to think of value in terms of an equation is as follows:

$$Value = Performance/Cost \tag{1}$$

A more sophisticated version of this algorithm is described as follows:

$$V_f(P, C, t)_{total} = \frac{\sum_{n=1}^{\infty} P_n \cdot \alpha}{\sum_{n=1}^{\infty} [C_n \cdot \alpha) + (t_n \cdot \alpha)]} \tag{2}$$

where $V$ is the Value, $P$ is the Performance, $t$ is the Time, $f$ is the Function, $C$ is the Cost, $\alpha$ is the Risk.

However, cost reduction is emphasized in practice and the possibility of insufficient review to determine structural safety and quality has been pointed out as a problem. When implementing VE, alternative design focusing on cost reduction has also been proposed because of the misconception that VE is a simple cost saving method. Even when an alternative design is presented, it is pointed out that the objective evaluation of function and cost is not performed [7]. In addition, there are various evaluation items and evaluation methods. Even if the subjects to be evaluated are the same, the evaluation results may differ depending on the evaluation items and evaluation methods [8]. The biggest problem of the existing methods used in practice is that the calculation ranges of the function and cost indices are varied according to the functional attribute and the cost attribute of the evaluation subject. Indexing of cost scores and function scores that make function and cost indices within the same range is regarded as a difficult problem [9]. The client should select the optimum design from original design or an alternative design using the function score, the cost score and the value score, and decide whether to apply it or not. However, because the calculation range of the function, cost and value scores differs for each evaluation subject, the client often has difficulty in selecting the optimum design and deciding whether or not to apply it. If the result of the decision rule is the same, the rule corresponding to the decision must be applied and the result must be calculated within the rule range [10]. This is because there is no difficulty in making a decision. In VE, when the calculation ranges of function score, cost score and value score of original design and alternative design are different, several decision-making rules are generated and a single decision-making rule that unifies the calculation range of the function score, cost score and value score can be helpful for efficiency and accuracy in making decisions. This study suggests a numerical model that can calculate the function score, cost score and value score in the same range to support the decision-making for selecting the optimum design of the client. It takes a lot of time and effort to make decisions in selecting the optimum design because you must compare the function score, the cost score and the value score of many different designs. The purpose of this study is for the client to improve the efficiency of decision-making in the process of selecting the optimum design in VE.

## 2. Literature Review

VE should be utilized to produce optimal buildings. It is also stated that for optimum design, concurrent engineering should be considered for manufacture and assembly, total quality management and life-cycle design [11]. The effect of VE could be doubled by considering the system in the engineering aspect and other factors separately. The original design and the alternative design were evaluated as economy-driven by analyzing the data of the designs from a VE competition held by the Korean government. In addition, it was found that the evaluation results also varied if the evaluation items and the evaluation methods were different for the same VE service. To improve these points, it was suggested that evaluation items were classified into upper, middle and lower level categories, and proposed a hierarchy of the evaluation items using the Analytic Hierarchy Process technique and a method of selecting an optimum design [8]. It was suggested that VE evaluation items should be classified into quantitative items and qualitative items. In order to compare precast concrete columns and steel columns, he classified the evaluation items into quantitative and qualitative elements, and calculated the weights of the evaluation items using the Analytic Hierarchy Process technique [12]. Then, he suggested a model for selecting the optimum design by weighted sum using a Fuzzy Set. The Interactive Value Management System, which is a Group Decision Support System for efficient VE, was

devloped. The Interactive Value Management System is designed to enable bidirectional data transmission using web-based IT technology and it makes the Group Decision Support System possible [13]. Previous studies have been focused on the development of functional evaluation methods using the Analytic Hierarchy Process, but there is insufficient research to integrate functions and costs.

## 3. Study Procedures and Methods

The study procedures and methods are as follows.

First, the decision-making method is analyzed for indexing the function score, the cost score and the value score. A suitable indexing method for the study is then selected. Second, the study model is designed using the indexing method selected above. Third, VE is the method of dividing the function score by the cost (hereinafter, Method 1), and the value score calculation method of the California Department of Transportation (hereinafter, Method 2), and the value score calculation method using the indexing model presented in this study (hereinafter, Method 3) are applied. Fourth, The value calculated by applying the existing method and the value calculated by applying this study model are compared and analyzed. Fifth, summarize the results of the study and present the limitations and future tasks of this study.

Figure 1 shows the procedures and methods of Method 1, Method 2, and Method 3.

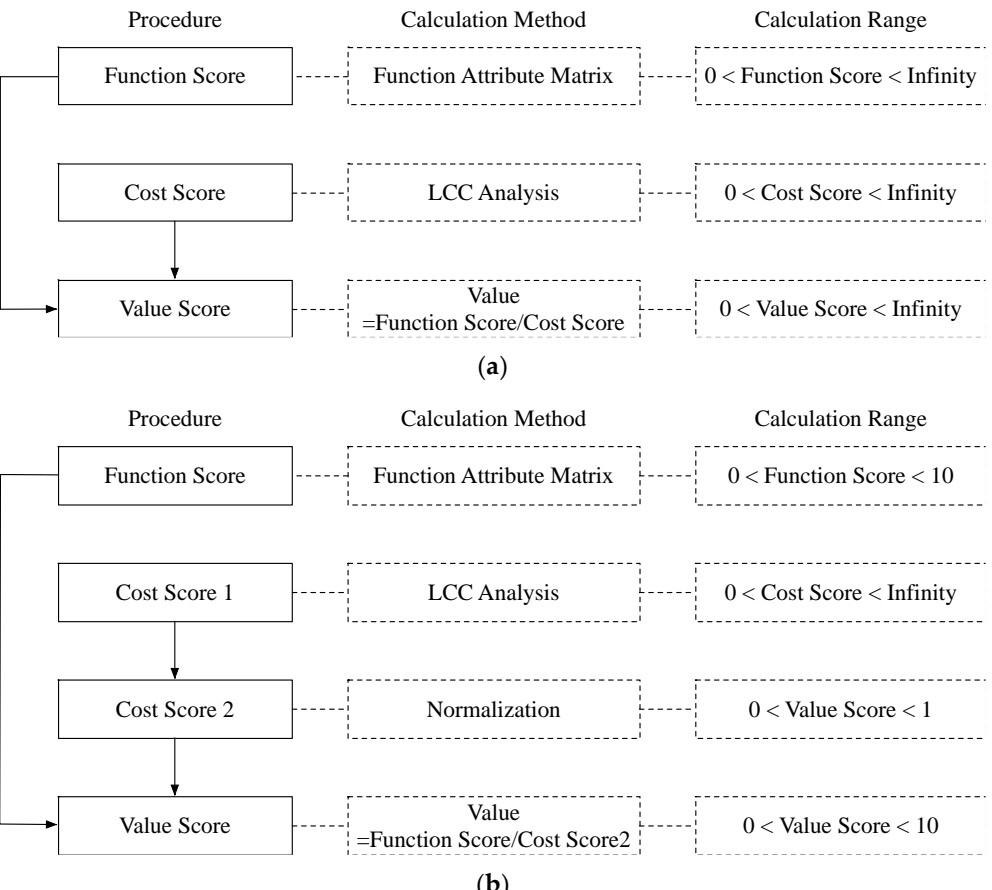

**Figure 1.** *Cont.*

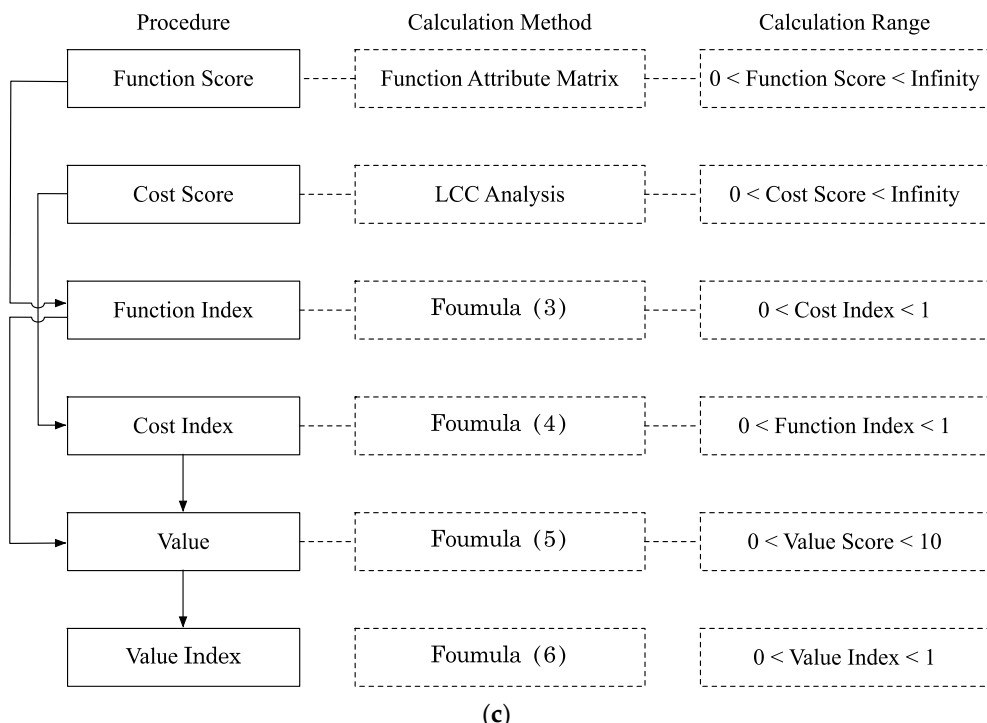

(c)

**Figure 1.** Procedures of (**a**) Method 1, (**b**) Method 2, and (**c**) Method 3.

## 4. Study Model

### 4.1. Selection of Indexing Method

It is important when making the decision to compare the function, cost and value scores of the original design and the alternative design presented through VE. Decision-making is divided into multi-objective decision-making and multi-attribute decision-making [14]. Although the architectural project includes both the characteristics of multi-objective decision-making and multi-attribute decision-making, VE, which is the subject of this study, evaluates a limited number of designs based on original designs and alternative designs. It also shares a common goal of building value enhancement and excludes other designs for optimum design. Table 1 summarizes the features of multi-objective decision-making and multi-attribute decision-making and the characteristics of VE mentioned above correspond to multi-attribute decision-making. In multi-attribute decision-making, different criteria must be converted to the same criteria according to attributes [15]. Converting and indexing function and cost scores into comparable scales can be done by (1) the linear transformation method which divides the maximum of each evaluation value by the remaining evaluation values, (2) the normalization method which uses the average value of the evaluation values, (3) the normalization method which uses the intermediate value of the evaluation values, and (4) a vector normalization method which divides the evaluation value into the norm of the evaluation values [1]. The first linear transformation method is a suitable method when the higher the evaluation value, the better the item and the lower the evaluation value, the better the items exist together. It is possible to rearrange different evaluation values with different preferences and convert them to the same preferences. However, the first linear transformation method is not suitable for indexing VE function scores and cost scores because it converts the lowest valued items to '0'. The second normalization method using the average value of the evaluation values assumes that the average value of the evaluation values is '0', and the third normalization method using the intermediate value of the evaluation values also assumes the intermediate value of the evaluation values is '0'. The normalization method using the average value and the normalization method using the intermediate value set the average value or the intermediate value as '0' and place the other evaluation values on the left and right. Therefore,

the normalization method using the average value and the normalization method using the intermediate value are not suitable for indexing the function score and the cost score because they have a negative value when the normalization value becomes smaller than the normalized value of '0'. The fourth vector normalization method sets the norm of the vector to '1' and calculates the rate of each vector. The vector normalization method is suitable for indexing function scores and cost scores because the evaluation values can be converted to a certain range $(0 < x < 1)$. Therefore, in this study, the vector normalization method is used to design this study's model to index function scores and cost scores.

**Table 1.** Features of multiple criteria decision-making.

| Classification | Multi-Objective Decision-Making | Multi-Attribute Decision-Making |
|---|---|---|
| Criteria | Purpose | Element |
| Alternative | Infinity | Finite |
| Purpose | Explicit | Implicit |
| Element | Implicit | Explicit |
| Constraint condition | Active | Inactive |
| Use | Design | Select/Assessment |

A vector is a directed line segment from the starting point x to the ending point y of the two-dimensional space and is defined as an ordered pair of two points on the coordinate system. Here, a vector has a size and a direction, and the vector of the same size and direction is called equivalent. On the coordinate system, there exist vectors with position information that are equivalent but have different starting points. However, if the starting points of all vectors move to the origin '0', they can be expressed as the only vector having position information by the ending points. The real number sequence column of n that determines the position of the end point is called the coordinate of the vector. In addition, the norm of the vector is defined as $\sqrt{a^2 + b^2 + c^2 +, \cdots, +n^2}$ when the coordinate system in which this vector is defined is the n -dimensional space *Rn* [16]. The vector normalization is obtained by dividing each column vector by the norm, and then by calculating the rate of each vector by seeing the defined norm as '1' [17]. Therefore, the function score and the cost score of the original design and the alternative design are defined as a vector of n-dimensional space, and the corresponding vector is divided into the norm and it can be indexed into a unit vector with a starting point of '0' and a maximum size of '1'.

*4.2. Indexing Model*

**(1) Function Index**

When the function score norm of original design and alternative design is $\| F \|, \| F \| = \sqrt{\sum_{i=1}^{n} F_i^2}$. Therefore, function index (hereinafter, *FI*), which is a normalized function score, can be calculated by dividing the function score *F* of the evaluation subject *i* by $\| F \|$. The indexing model for the *FI* calculation is shown in the following Equation (3):

$$FI_i = \frac{F_i}{\| F \|} = \frac{F_i}{\sqrt{\sum_{i=1}^{n} F_i^2}} \tag{3}$$

where $FI_i$ is the function index of the evaluation subject *i*, $F_i$ is the function score of the evaluation subject *i*, *F* is the function, and *i* is the evaluation subject $(i = \{1, \cdots, n\})$. Function scores of the original design and the alternative design calculated using the value matrix are substituted into Equation (3) and indexed. Since the function index is calculated by dividing the function score *F* of the evaluation subject *i* by $\| F \|$, the calculation range is $0 < FI < 1$.

**(2) Cost Index**

If the LCC norm of the original design and the alternative design is defined as $\| LCC \|$, then $\| LCC \| = \sqrt{\sum_{i=1}^{n} LCC_i^2}$. Therefore, cost index (hereinafter, *CI*), which is a normalized LCC, can be calculated by dividing the LCC of the subject $i$ by $\| LCC \|$. The indexing model for *CI* calculation is shown in the following Equation (4):

$$CI_i = \frac{LCC_i}{\| LCC \|} = \frac{LCC_i}{\sqrt{\sum_{i=1}^{n} LCC_i^2}} \tag{4}$$

where $CI_i$ is the cost index of the evaluation subject $i$, $C_i$ is the LCC of the evaluation subject $i$, $C$ is the LCC and $i$ is the evaluation subject ($i = \{1, \cdots, n\}$). Cost scores of the original design and the alternative design calculated through LCC analysis are substituted into the Equation (4). Since the cost index is calculated by dividing the LCC of the evaluation subject $i$ by $\| LCC \|$, the calculation range is $0 < CI < 1$, as is the calculation range of the function index.

**(3)　Value Index**

In the case of using the VE theory to calculate the rate of function and the cost in calculating the value index, the larger the difference between the function index and the cost index, the larger the unit of the value index. Assuming that the function index is fixed, the cost index of the design with a smaller LCC value becomes closer to '0' and the value index becomes infinite as the LCC difference between the original design and the alternative design increases. For example, if the function index is 0.995 and the cost index is 0.001, the value index will be 995. The function index is equal to 0.995 and when the cost index is 0.0001, the value index is 9950. Therefore, in this study, the function index is calculated as seen in Equation (5), and is divided by the cost index, and the value is calculated and indexed using the vector normalization method.

Thus, the value index of this study model follows two stages explained below.

First, calculate the value score by dividing the function index by the cost index as in Equation (5):

$$V_i = \frac{FI_i}{CI_i} \tag{5}$$

where $V_i$ is the value of the evaluation subject $i$, $FI_i$ is the function index of the evaluation subject $i$, $CI_i$ is the cost index of the evaluation subject $i$ and $i$ is the evaluation subject ($i = \{1, \cdots, n\}$).

Next, if the value norm of the original design and the alternative design is defined as $\| V \|$, then $\| V \| = \sqrt{\sum_{i=1}^{n} V_i^2}$. Therefore, value index (hereinafter, VI), which is a normalized value, can be calculated by dividing the value of the evaluation subject $i$ by $\| V \|$. The index model for VI calculation is shown in the following Equation (6):

$$VI_i = \frac{V_i}{\sqrt{\sum_{i=1}^{n} V_i^2}} \tag{6}$$

where $VI_i$ is the value index of the evaluation subject $i$, $V_i$ is the value of the evaluation subject $i$, and $i$ is the evaluation subject ($i = \{1, \cdots, n\}$). Since the value index is calculated by dividing the value of the evaluation subject $i$ by $\| V \|$, the calculation range is $0 < VI < 1$.

## 5. Case Study

The case study focuses on the computer room floor design of the office building. Method 1, Method 2, and Method 3 were applied to the VE cases, and the calculated values were compared. As shown in Figure 2, the original design was designed as a vibration reduction and laminate floor for both access floor and office automation floor (hereinafter,

OA floor). The alternative design was designed as a conductive tile for the access floor and the art deco tile for the OA floor. This study applied the Method 1, Method 2, and Method 3 of this study to the original design and the alternative design, and compared calculation results of the function score, the cost score, the value score, the function index, the cost index and the value index.

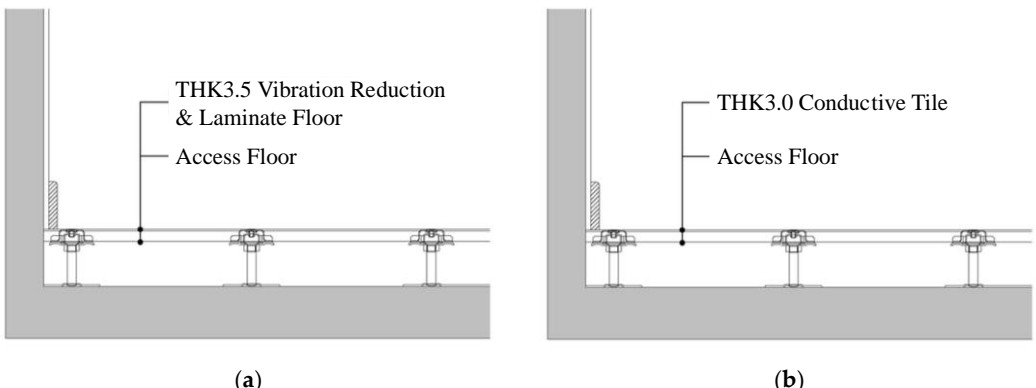

(**a**)          (**b**)

**Figure 2.** Case study subject. (**a**) Original Design; (**b**) Alternative Design.

*5.1. Calculating Function Scores and Cost Score*

**(1)   Calculating Function Scores**

A value matrix is a method of measuring the functions of the original design and the alternative design using functional values. As shown in Table 2, the functional evaluation items analyzed through the job plan are 'Convenience of outside circulation', 'Usage of environmental friendly materials', 'Durability (Endurance limit)', 'Ease of maintenance' and 'Hazard Prevention (Fire and disaster)'. First, using the weight calculation method of the value matrix as shown in Table 2, the weight was calculated by analyzing the relative dominance among the five functional evaluation items. 'Convenience of outside circulation' is analyzed as the most important functional evaluation item, followed by 'Durability (Endurance limit)', 'Ease of maintenance', 'Hazard Prevention (Fire and disaster)' and 'Usage of environmental-friendly materials'.

**Table 2.** Function Attribute Matrix using Analytic Hierarchy Process Paired Comparison.

|  | A: Convenience of Outside Circulation | B: Durability (Endurance Limit) | C: Ease of Maintenance | D: Hazard Prevention (Fire and Disaster) | E: Usage of Environmental-Friendly Materials | Weight |
|---|---|---|---|---|---|---|
| A: Convenience of outside circulation | 1 | 4 | 3 | 3 | 4 | 41.5% |
| B: Durability (Endurance limit) | 0.250 | 1 | 1 | 5 | 5 | 21.9% |
| C: Ease of maintenance | 0.333 | 1.000 | 1 | 4 | 5 | 21.2% |
| D: Hazard Prevention (Fire and disaster) | 0.333 | 0.200 | 0.250 | 1 | 4 | 10.3% |
| E: Usage of environmental-friendly materials | 0.250 | 0.200 | 0.200 | 0.250 | 1 | 5.1% |
| Subtotal | 2.17 | 6.40 | 5.45 | 13.25 | 19.00 | 100.0% |

An evaluation of function grades of the original design and the alternative design was performed using a 10-point scale. The function grade of the original design was set to '5', which is the intermediate value of the evaluation scale. Based on the original design, the grade of the alternative design was compared and evaluated to calculate the function grade of the detailed items in the alternative design. The function score of the detailed

evaluation items is calculated by multiplying the weight and the function grade of the detailed evaluation items. The function score of the detailed evaluation items was added up to calculate the final function score. The function score of the original design was calculated as 5.000 and the function score of the alternative design was calculated as 5.959. Therefore, it is analyzed that the alternative design is superior to the original design in terms of functionality. Table 3 shows the function score calculation data calculated using the value matrix.

**Table 3.** Function score calculation data.

| Names of Items | Weight | Original Design | | Alternative Design | |
|---|---|---|---|---|---|
| | | Grade | Score | Grade | Score |
| Convenience of outside circulation | 41.5% | 5 | 2.074 | 5 | 2.074 |
| Durability (Endurance limit) | 21.9% | 5 | 1.096 | 7 | 1.534 |
| Ease of maintenance | 21.2% | 5 | 1.059 | 6 | 1.270 |
| Hazard Prevention (Fire and disaster) | 10.3% | 5 | 0.517 | 7 | 0.724 |
| Usage of environmental-friendly materials | 5.1% | 5 | 0.255 | 7 | 0.357 |
| Total | 100 | - | 5.000 | - | 5.959 |

**(2)　Calculating Cost Score**

In VE, the cost evaluation consists of initial cost and running cost and analyzes LCC. The initial cost, the construction cost, is the direct construction cost. Direct construction cost consists of material cost, labor cost and expenses. The LCC analysis results are used as cost scores. In VE, the initial cost is the construction cost, and the construction stage is set as the present point in the LCC analysis. Therefore, the construction cost itself becomes the present value. The running cost is the future cost of maintenance in the process of using the building after completing construction. It is divided into the cost that is repeated every year (recurring cost) and costs which are not required to be repeated every year but at a certain time (non-recurring cost). The running cost is equivalent to the construction cost at the same point in time when using the present value method. At this time, in order to convert future costs to present value, it is necessary to consider the change in value of money with time. The real discount rate is the rate of change in the cost value over time. The real discount rate is calculated using the nominal discount rate and the inflation rate as shown in Equation (7) below. Recurring cost is converted into present value using Equation (8), and non-recurring cost is converted into present value using Equation (9).

$$i_r = \{(1 + i_n)/(1 + f)\} - 1 \tag{7}$$

$$RPV = \{(1 + i_r)^m - 1\}/\{i \times (1 + i_r)^m\} \times RC \tag{8}$$

$$NPV = \{i \times (1 + i_r)^n\} \times NC \tag{9}$$

where $RPV$ is the present value of the recurring cost, $NPV$ is the present value of the non-recurring cost, $RC$ is the recurring cost, $NC$ is the non-recurring cost, $m$ is the period of occurrence of recurring cost, $n$ is the time of occurrence of non-recurring cost, $i_r$ is the real discount rate, $i_n$ is the nominal discount rate, and $f$ is the inflation rate.

As shown in Table 4, the initial cost and the running cost of the original design and the alternative design were compared and analyzed. The initial construction cost of vibration reduction and laminate flooring of the original design was ₩36,666,000. The initial cost of conductive tile and art deco tile, which are the components of the alternative design, were ₩5,990,906 and ₩13,599,762, respectively. According to the client's request, 50 years are applied for the life cycle of the building and the real discount rate is 3%. The LCC was then analyzed. The LCC of the original design was ₩70,615,530 and the LCC of the alternative design was ₩54,658,790. Therefore, it is analyzed that when changing to the alternative design, 29% was saved.

Table 4. Results of LCC analysis.

| Composition | Classification | Quantity (m²) | Unit Price (₩) | Initial Cost (₩) | Cycle of Maintenance (Year) | Rate of Repairing Level (%) | LCC Analysis | | Total LCC (₩) | Cost Score |
|---|---|---|---|---|---|---|---|---|---|---|
| | | | | | | | Present Worth | LCC (₩) | | |
| Original Design | Vibration reduction and laminate Floor | 873 | 42,000 | 36,666,000 | 7 | 15 | 0.813092 | 4,471,925 | | 0.564 |
| | | | | | 14 | 15 | 0.661118 | 3,636,083 | | |
| | | | | | 21 | 15 | 0.553676 | 3,045,163 | | |
| | | | | | 25 | 100 | 0.477606 | 17,511,902 | | |
| | | | | | 32 | 15 | 0.388337 | 2,135,815 | | |
| | | | | | 39 | 15 | 0.315754 | 1,736,615 | | |
| | | | | | 46 | 15 | 0.256737 | 1,412,028 | | |
| | Total | | | 36,666,000 | | | | 33,949,530 | 70,615,530 | |
| Alternative Design | Conductive tile | 191 | 31,366 | 5,990,906 | 9 | 9 | 0.766417 | 413,238 | | 0.436 |
| | | | | | 18 | 9 | 0.587395 | 316,713 | | |
| | | | | | 19 | 100 | 0.570286 | 3,416,530 | | |
| | | | | | 28 | 9 | 0.437077 | 235,664 | | |
| | | | | | 37 | 9 | 0.334983 | 180,617 | | |
| | | | | | 38 | 100 | 0.325226 | 1,948,398 | | |
| | | | | | 47 | 9 | 0.249259 | 134,396 | | |
| | Art deco tile | 682 | 19,941 | 13,599,762 | 7 | 15 | 0.813092 | 1,658,679 | | |
| | | | | | 12 | 100 | 0.70138 | 9,538,601 | | |
| | | | | | 19 | 15 | 0.570286 | 1,163,363 | | |
| | | | | | 24 | 100 | 0.491934 | 6,690,185 | | |
| | | | | | 31 | 15 | 0.399987 | 815,959 | | |
| | | | | | 36 | 100 | 0.345032 | 4,692,353 | | |
| | | | | | 43 | 15 | 0.280543 | 572,298 | | |
| | | | | | 48 | 100 | 0.241999 | 3,291,129 | | |
| | Total | | | 19,590,668 | | | | 35,068,122 | 54,658,790 | |

### 5.2. Applying Method 1

The function score calculated using the Method 1 was divided by the cost score and then calculated by the LCC analysis to determine the value score. The function score of the original design was 5.000 and the function score of the alternative design was 5.959. Meanwhile, the cost score was ₩70,615,530 for the original design and ₩54,658,790 for the alternative design. When these scores were substituted into the equation '*Value* = Function/Cost', the equation for VE value scores, the value score of the original design was 0.0000000708 and the value score of the alternative design was 0.0000001090 (Table 5).

**Table 5.** Evaluation data 1.

|  | Function Score | Cost | Value Score | Value Enhancement Rate |
|---|---|---|---|---|
| Original Design | 5.000 | 70,615,530 | 5.000/70,615,530 = 0.0000000708 | - |
| Alternative Design | 5.959 | 54,658,790 | 5.959/54,658,790 = 0.0000001090 | +54% |

As described above, there is a large difference in the calculation range of function scores, cost scores and value scores.

### 5.3. Applying Method 2

In order to overcome the disadvantages of Method 1, Method 2 changes the calculation range of the cost score. The sum of the cost of the original design and the alternative design is set to '1', and the cost ratio of the original design and the alternative design is used as the cost score. The function score is used as it is, and the cost score is changed to 0.564 for the original design and 0.436 for the alternative design. Therefore, as shown in Table 6, the value score of the original design is 8.870 and the function score of the alternative design is 13.657.

**Table 6.** Evaluation data 2.

|  | Function Score | Cost Score | Value Score | Value Enhancement Rate |
|---|---|---|---|---|
| Original Design | 5.000 | 70,615,530/(70,615,530 + 54,658,790) = 0.564 | 5.000/0.564 = 8.870 | - |
| Alternative Design | 5.959 | 54,658,790/(70,615,530 + 54,658,790) = 0.436 | 5.959/0.436 = 13.657 | +54% |

### 5.4. Applying Method 3

**(1)　Calculating Function Index**

The function score of the original design and the alternative design calculated using the value matrix were substituted into Equation (3), which is the indexing model of the function scores presented in this study. Then, the function index of the original design and the function index of the alternative design were calculated. As shown in Table 7, the function index of the original design was 0.666 and the function index of the alternative design was 0.794.

**Table 7.** Function index calculation data.

|  | Function Score | Function Index |
|---|---|---|
| Original Design | 5.000 | $\frac{5.000}{\sqrt{5.000^2 + 5.959^2}} = 0.666$ |
| Alternative Design | 5.959 | $\frac{5.959}{\sqrt{5.000^2 + 5.959^2}} = 0.794$ |

**(2)　Calculating Cost Index**

The cost of the original design and the alternative design calculated using the value matrix were substituted into Equation (4), which is the indexing model of the cost scores presented in this study. Then the cost index of the original design and the cost index of the alternative design were calculated. As shown in Table 8, the function index of the original design was 0.791 and the function index of the alternative design was 0.612.

**Table 8.** Cost index calculation data.

| | Cost | Cost Index |
|---|---|---|
| Original Design | 70,615,530 | $\frac{70,615,530}{\sqrt{70,615,530^2+54,658,790^2}} = 0.791$ |
| Alternative Design | 54,658,790 | $\frac{54,658,790}{\sqrt{70,615,530^2+54,658,790^2}} = 0.612$ |

**(3)    Calculating Value Index**

Using Equations (3) and (4), the values of the original design and the alternative design were calculated by substituting the calculated function index and the cost index of the original design and the alternative design into Equation (5). As shown in Table 9, the value score of the original design was 0.843 and the value index of the alternative design was 1.297.

**Table 9.** Value Score calculation data.

| | Function Index | Cost Index | Value Score | Value Enhancement Rate |
|---|---|---|---|---|
| Original Design | 0.666 | 0.791 | 0.666/0.791 = 0.843 | - |
| Alternative Design | 0.794 | 0.612 | 0.794/0.612 = 1.297 | +54% |

Next, the values of the original design and the alternative design were calculated by substituting the previously calculated values of the original design and alternative design into Equation (6). As shown in Table 10, the value index of the original design was 0.545 and the value index of the alternative design was 0.838.

**Table 10.** Value index calculation data.

| | Value | Value Index | Value Enhancement Rate |
|---|---|---|---|
| Original Design | 0.843 | $\frac{0.843}{\sqrt{0.843^2+1.297^2}} = 0.545$ | - |
| Alternative Design | 1.297 | $\frac{1.297}{\sqrt{0.843^2+1.297^2}} = 0.838$ | +54% |

### 5.5. Result Comparison

The results calculated using Method 1, the results calculated using Method 2, and the results calculated using Method 3 were compared with each other. The function score of the original design calculated using Method 1 was 5000 and the function score of the alternative design was 5.959. In addition, the cost of the original design was 54,658,790 and the cost of the alternative design was 70,615,530. Thus, the gap of the calculation range of the function score and the cost is large in Value Matrix 1. Since Value Matrix 1 uses a weight with a sum of 100 when calculating the function score and a 10-point scale for the functional grade comparison, a function score with a minimum of 1 and a maximum of 10 can be calculated. The cost score can be calculated from '1' to infinity, depending on the cost attributes of the original design and the alternative design. Additionally, when using the existing method, which is Method 1, the value score of the original design is 0.0000000708 and the value score of the alternative design is 0.0000001090. Thus, the gap

of the calculation range of the function score and the cost score is large (Figure 3a). When using Method 2, the calculation range of the cost score was reduced to 0.564 for the original design and 0.436 for the alternative design. However, the value score is 8.870 for the original design and 13.657 for the alternative design, and the gap of the calculation range of the function score and the cost score is large (Figure 3b). When applying the Method 3 proposed in this study, the function index of the original design is 0.666 and the function index of the alternative design is 0.794. In addition, the cost index of the original design is 0.791 and the cost index of the alternative design is 0.612. The value index of the original design is 0.545 and the value index of the alternative design is 0.838. When the indexing model presented in this study was applied, the function index, the cost index and the value index were all calculated in the range of $0 < x < 1$ (Figure 3c). The value enhancement rate using Method 1, the value enhancement rate using Method 2, and the value enhancement rate using the Method 3 presented in this study were compared. The value enhancement rate of Method 1, Method 2, and Method 3 presented in this study was equal to + 54% (Figure 3d).

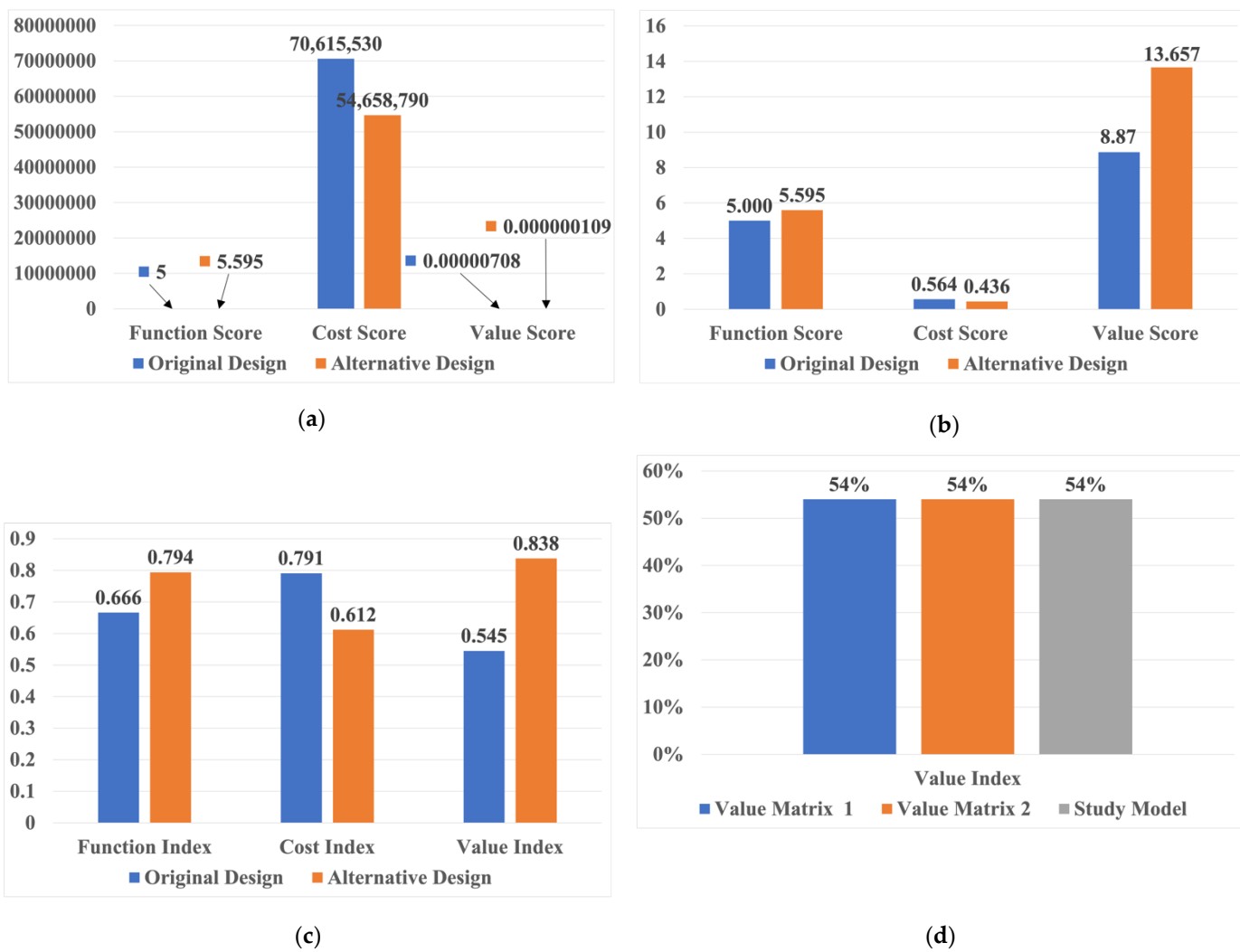

**Figure 3.** Result comparison data. (**a**) Method 1; (**b**) Method 2; (**c**) Method 3; (**d**) Gap in value enhancement rate between existing Method 1, Method 2 and Method 3.

## 6. Conclusions

The purpose of this study is to index the function score, the cost score, and the value score in VE, which is used to improve building value. The main results are as follows.

Decision-making techniques were analyzed to select the indexing model design method. As a result, the evaluation of the original design and the alternative design and the selection of the optimum design in VE were analyzed as belonging to the multiple-attribute decision-making. In the multiple-attribute decision-making methodology, since the target values to be normalized can be converted into a certain range, the vector normalization method is analyzed in a manner suitable for indexing the function score, the cost score and the value score, which is the purpose of this study. The function index is designed to be calculated by dividing the function score $F$ of the evaluation subject $i$ by $\| F \|$. The cost index is designed to be calculated by dividing the LCC of the evaluation subject $i$ by $\| LCC \|$. The value index calculation is designed in two stages. The reason for designing the value index in two stages is that if the value index is calculated by dividing the function index into the cost index, the value index becomes infinite as the cost index is lowered. The value index can be calculated by first dividing the function index into the cost index, and then the value index can be calculated using the calculated value. The value index is designed to be calculated by dividing the value of the evaluation subject $i$ by $\| V \|$. Method 1 (to calculate the value score by dividing the function score by the cost score), Method 2 (of the California Department of Transportation to reduce the calculation range of the function score and the cost score), and Method 3 (presented in this study) were applied to the office building computer room floor design VE case and the results were compared. When using Method 1, the function score of the original design was 5.000 and the function score of the alternative design was 5.595. The cost of the original design through the LCC analysis was 54,658,790 and the cost of the alternative design through the LCC analysis was 70,615,530. The value score calculated by dividing the function score by the cost score was 0.0000000708 in the original design and 0.0000001090 in the alternative design. Thus, the gap in the calculation range of the function score, the cost score and the value score was very large. Next, in the case of Method 2, the cost score of the original design was converted to 0.564 and the cost score of the alternative design was converted to 0.436. The value score of the original design was 8.870 and the value score of the alternative design was 13.657. There was a large gap in the calculation range between the function score and the cost score. The calculation results of the function index, the cost index and the value index when applying the Method 3 presented in this study were as follows. The function index of the original design was 0.666 and the function index of the alternative design was 0.794. The cost index of the original design was 0.791 and the cost index of the alternative design was 0.612. Moreover, the value index of the original design was 0.545 and the value index of the alternative design was 0.838. Therefore, it can be confirmed that the function index, the cost index and the value index are all calculated in the same range $(0 < x < 1)$.

This study analyzed the change in value enhancement rate using Method 1, Method 2, and the Method 3 presented in this study. The value enhancement rate of Method 1, Method 2, and Method 3 was equal to + 54%. Therefore, the value enhancement rate when applying the Method 3 presented in this study is equal to the value enhancement rate when applying Method 1 and Method 2, and it is interpreted that the attributes of the function score and the cost score in the indexing process are not modified. With the recent development of a variety of building materials, designs are becoming more diverse. This means that there are more variables in design optimization and optimum design selection. Furthermore, due to the development of various building materials and the diversification of designs accordingly, many alternatives have been presented through VE and a lot of time and effort has been required for decision-making when selecting the optimum design among the proposed alternatives. This study designed a model to index function scores, cost scores and value scores in VE, and confirmed the consistency and practical applicability of the indexing model through case studies.

The value is analyzed by substituting the function score and the cost score to the indexing model presented in this study. Since the value index is calculated using the analyzed value, it is possible to shorten the time required for decision-making to select the optimum design among the designs. It is also possible to make a coherent evaluation of the

design with various functional and cost characteristics, and thus it can be used for rational decision-making when selecting the optimum design. However, the model proposed in this paper can calculate the cost index and function index only if there is more than one alternative design. Furthermore, as the number of alternative designs increases, the cost index interval and the function index interval between the alternative designs decreases. Then, decision-making becomes difficult. In the case study, the function score, cost score, and value score could be indexed into values greater than 0 and less than 1 using this study model. However, as the number the decimal places of the exponential value increases, the possibility of error in decision making may increase. Therefore, when the function score, cost score and value score are indexed, a follow-up study is needed to reduce the number of decimal places of the calculated index value.

**Funding:** This research received no external funding.

**Institutional Review Board Statement:** Not applicable.

**Informed Consent Statement:** Not applicable.

**Data Availability Statement:** Data are contained within the article.

**Acknowledgments:** This research was supported by the Basic Science Research Program through the National Research Foundation of Korea (NRF) funded by the Ministry of Education (2020R1I1A3A040 36790).

**Conflicts of Interest:** The authors declare no conflict of interest.

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
