# Peer review of "Indexing Model Based on Vector Normalization Available for Value Engineering in Building Materials"

_applsci, doi:10.3390/app11209515_

Round 1

Reviewer 1 Report

The abstract ist a bit messy. An additional sentence introducing the method of value engineering might be helpful; Also terms such as functions score ... might be introduced, otherwise the abstract feels like terminology-bingo;

You should mention your case study in the abstract.

In general, it is recommended to shorten sentences and to increase understandibility, which is not very emphasized throughout the introduction;

Why you use the [] reference version and then mention two authors in a different style (Roman, S. and Stefanowski, J.) remains a mystery.

Even if you mention the author of a study as a sentence starter, it does not make any sense to use a comma and then the first initial. Instead of Omigbudon, A ... ist should Omigbudon [reference no in the reference section]...

What is OA floor? you should mention abbreviations the first time you use the,

 Is table 2 now on one of the versions, or generally? this is messy again.

In your results you show that all three methods have the same 54% value enhancement.... so what does this tell us?

Does it really make sense to compare to absolute numbers, such as done here? The value score calculated by dividing the function score by the cost score
397 was 0.0000000708 in the original design and 0.0000001090 in the alternative design.

This is - sorry to say so - bad scientific style. Such numbers are not to be understood meaninfully. u have to count the zeros after the floating point to be able to compare this two numbers.

I believe that this paper should be devlared as case study paper rather than as method comparison, given that 1(!!!!) case study is used, and it might not be ensured that your results are generic.

I plead for major revision. In general it might be a interesting article, but both the presentation and the way information is transported need strong revision

Author Response

I appreciate the review.

I attached the reply to the review report.

Reviewer 2 Report

The paper presents an interesting idea that can be a real help in selecting the optimum solutions for buildings elements. But...

  • in the abstract, the author refers to clients who are confused in the decision-making process of selecting the optimum design, but the entire study is hard to understand and the proposed model can not be used by any "client" without solid knowlidge regarding the value engineering, thus reducing the interest of many actors within the constructions sector; 
  • title refers to "building materials", but the study is about an entire "optimum design" which means that a decision can not be made based only on choosing the building materials, but also taking into consideration the technologies used that implies costs for manpower, machinery, transportation, etc; 
  • introduction and the literature review can be improved in order to justify the necessity of the study and its novelty;
  • a better description of those three methods should be made, highlighting the pros and cons of each one, in this way, again, the novelty and usefulness of the present study being emphased;
  • if the final results are similar, why method 3 should be used instead of the first ones that seems much easier?
  • conclusions are not answering to the above question; more, there could be a misunderstanding regarding the number of alternative designs: the author says that the proposed model can be used only if there are more than one alternative designs considered, but an increased number of alternative designs makes decision-making more difficult; this aspect should be rephrased as it can be understood as a contradiction (although it is reason for the proposed model);
  • there are used some abbreviation without being explained before: OA floor, AHP, MADM, etc;
  • some references can not be verified / accessed;
  • slightly revise the English language;

Author Response

(The authors gave the same response as above.)

Reviewer 3 Report

The article is interesting, although the subject matter is apparently hermetic, as the author cited only 19 items of literature, including two websites (one of which is no longer available) and four thesis (in Korean), while one of the items is not quoted in the text. The method presented is described in an understandable way, although it is difficult to determine where some of the values in Tables 2 and 4 came from.

The approach to the calculation of VE presented by the author is not particularly innovative or inventive. But because it is useful and original, it is worth presenting it to a wider circle of readers. Before this happens, however, certain amendments are necessary, which are listed below in the points:

  1. Citations should be restructured and the following corrections should be made:
    • The items cited should be arranged in the order in which they appear in the text, and not alphabetically.
    • Item [1] is no longer available. It should be removed.
    • Item [8] provides forenames instead of surnames. It should be: Kelly J. and Male S.
    • Item [12] gives the first author's name instead of his last name. It should be: Slowinski R.
    • In item [18], the first name is given instead of the last author. It should be: Kelly J.
    • Item [19] has also been published in paperback version, so it does not need to be cited as a website. But if the author thinks that this form is more appropriate, please shorten the link to https://natureofcode.com/book/ .
    • Item [7] is not referred to in the text. It should be cited or deleted.
    • Please indicate next to the literature items that are in a language other than English, in what language they are written.
  2. In line 214, in the formula, there should be V instead of LCC.
  3. In line 220, there should be VI instead of CI.
  4. Use parentheses correctly in formula (7).
  5. In Table 4 the descriptions of the first five columns should be adjusted.
  6. Please remove the redundant zeroes after the dot on the Y axis of the graphs in Figures 3b and 3c.

Author Response

(The authors gave the same response as above.)

Round 2

Reviewer 1 Report

Dear author,

I think you made steps forward,

however some of the answers are not really convincing. Keeping up several "0" in an evalutation value form seems not very fail-safe, indeed its error prone. I have my doubt that this is really the last word in application of the method.

The language is still somehow improvable.

e.g. VE should be utilized to produce optimum buildings [11]. No, it should be "optimal" buildings. And its an euphemism in it self, as there are really no optimal buildings, but building designs that are close to a Pareto-Front in self-defined optimae.

Moreover, I still believe that your example does not really lead to the conclusion that one needs the method of VE.

Be it as it may, I plead for accept after minor improvement. 

Author Response

I appreciate the review.

The points you pointed out have been modified/complemented as follows.

  1. however some of the answers are not really convincing. Keeping up several "0" in an evalutation value form seems not very fail-safe, indeed its error prone. I have my doubt that this is really the last word in application of the method.
  • Author’s reply:

It has been complemented as follows.

Reflecting your comments, I added the need for follow-up study to the conclusion.

  1. The language is still somehow improvable.

e.g. VE should be utilized to produce optimum buildings [11]. No, it should be "optimal" buildings. And its an euphemism in it self, as there are really no optimal buildings, but building designs that are close to a Pareto-Front in self-defined optimae.

  • Author’s reply:

Reflecting the reviewer’s comments, “optimum building” has been changed to “optimal building”. However, “optimum” and “optimal” are synonyms and are not semantically or grammatically incorrect in the manuscript. The manuscript was proofread by a native speaker, and a “Certificate of English Proofreading” was also attached.

Thank you for your interest in this manuscript.

Reviewer 2 Report

The author revised the paper according to some comments, but this revision was made mostly on the form and less on the content. 

My previous observations were not addressed entirely, so, I consider that the paper still needs a major revision in terms of presentation, better description of the methods used, emphasizing the novelty of this study, etc.

Author Response

Thank you for your interest in this manuscript.

I attached the reply to the review report.

Round 3

Reviewer 2 Report

The paper was improved by the author. There are still some issues previously mentioned, but, in general, I agree with the author's point of view. In this regard, I recommend to publish the paper as it is now.